# IgM Antiphospholipid Antibodies in Antiphospholipid Syndrome: Prevalence, Clinical Associations, and Diagnostic Implications—A Scoping Review

**DOI:** 10.3390/jcm14207164

**Published:** 2025-10-11

**Authors:** Monika Očková, Ariadna Anunciación-Llunell, Catalina Andrada, Enrique Esteve-Valverde, Francesc Miró-Mur, Jaume Alijotas-Reig

**Affiliations:** 1Systemic Autoimmune Diseases Research Unit, Vall d’Hebron Institut de Recerca (VHIR), Passeig Vall d’Hebron 119-129, 08035 Barcelona, Spain; monika.ockova@vhir.org (M.O.); ariadna.anunciacion@vhir.org (A.A.-L.);; 2Department of Internal Medicine, Hospital Sant Camil, Consorci Sanitari del Garraf, 08810 Barcelona, Spain; 3Department of Internal Medicine, Hospital Universitari Vall d’Hebron (HUVH), Passeig Vall d’Hebron 119-129, 08035 Barcelona, Spain; 4Department of Medicine, Universitat Autònoma de Barcelona (UAB), Vall d’Hebron Hospital Campus, Passeig Vall d’Hebron 119-129, 08035 Barcelona, Spain

**Keywords:** antiphospholipid syndrome, antiphospholipid antibodies, diagnosis, prevalence, obstetric manifestations, thrombotic manifestations, non-criteria aPL

## Abstract

**Background**: IgM antiphospholipid antibodies (aPL) were de-emphasised in the 2023 ACR/EULAR criteria, yet their precise clinical significance remains uncertain. **Methods**: A rapid scoping review of PubMed (January 2000–June 2025) identified original human studies reporting IgM aCL, aβ_2_GPI, or aPS/PT prevalence or outcomes; 40 studies met the eligibility criteria. Prevalence and odds ratios (ORs) of clinical associations were extracted. **Results**: IgM aPL are common across APS phenotypes. Obstetric cohorts showed aCL-IgM prevalences of 3–82%, often equal to or exceeding those of IgG, while aβ_2_GPI-IgM reached a prevalence of 2–63%. In mixed thrombotic–obstetric cohorts, aPS/PT-IgM was the most frequent isotype (31–79%). Purely thrombotic studies still reported 0–59% aβ_2_GPI-IgM, with PS/PT-IgM at 55% and 62% in two large series. Significant outcome signals from clinical associations of IgM aPL were inconsistent but noteworthy in (i) pregnancy loss for high-titre aCL, aβ_2_GPI, and aPS/PT, (ii) thrombosis driven by aPS/PT and (iii) organ-specific arterial events (retinal thrombosis and stroke) in isolated IgM phenotypes. **Conclusions**: The role of aPL-IgM remains uncertain. The findings advocate for a nuanced approach to IgM interpretation, supporting its reconsideration in specific clinical settings and emphasising the significance of ongoing research into the mechanistic and prognostic utility of IgM aPL.

## 1. Introduction

Antiphospholipid syndrome (APS) is a rare systemic thromboinflammatory disorder characterised by the persistent presence of antiphospholipid antibodies (aPL). Its clinical features include thrombosis (TAPS), obstetric morbidity (OAPS), and neurological or haematological complications [1,2,3]. These include venous thromboembolism, arterial thrombosis, microvascular diseases, thrombocytopenia, cardiac valve affections, and recurrent miscarriages, foetal death, and placental vasculopathies [4,5]. APS incidence is estimated to be 40–50 per 100,000 inhabitants, but these estimates are highly study-dependent [6,7]. The pathological aPL, currently recognised to be associated with APS, known as ‘criteria aPL’, include anti-cardiolipin (aCL), anti-β2 glycoprotein I (aβ_2_GPI), and lupus anticoagulant (LA).

For patients to be classified as having APS, they must meet both clinical criteria (exhibiting APS clinical symptoms) and laboratory criteria (being persistently positive for any of the aPL on at least two occasions, with a minimum interval of more than 12 weeks between tests) [8,9]. Patients without any other underlying autoimmune diseases are considered to have primary APS (PAPS). In contrast, those with systemic lupus erythematosus (SLE) or other autoimmune disorders are regarded as secondary APS (SAPS). Risk factors in APS are the presence of LA, double/triple positivity in high titres, concomitant SLE, classical cardiovascular risk factors such as hypertension, dyslipidaemia, obesity, smoking, and diabetes [10,11].

From 2006, APS classification was carried out according to the revised Sapporo criteria, also known as the Sydney criteria, which were later updated to the new ACR/EULAR criteria in 2023 [8,9] (Figure 1). New criteria sum-weighted scores per domains, and patients are classified as APS when their scores per clinical and laboratory domains are ≥3 points. aPL positivity needs to occur within a timeframe of 3 years after the event. In the Sydney criteria, both IgM and IgG isotypes of aCL and aβ_2_GPI were recognised as sound diagnostic predictors and were selected as laboratory classification criteria. For aCL, positivity was defined as titres exceeding 40 units or the 99th percentile, whereas for aβ_2_GPI, only titres above the 99th percentile were considered. In the 2023 revision of the criteria, isolated IgM positivity is no longer sufficient to meet the laboratory criterion; the presence of the IgG isotype or LA positivity is required for classification. Moreover, from 2023 onwards, the 99th percentile is no longer accepted as an appropriate aPL cut-off, and a uniform 40-unit cut-off value has been adopted to enhance the specificity of APS classification. Currently, diagnostic criteria for APS do not exist, but there are efforts to bridge classification and diagnosis [12,13].

IgM and IgG aPL are naturally occurring antibodies that are present in the otherwise healthy general population [7,14]. The prevalence of aPL in the general population is <10% for aCL and <5% for aβ_2_GPI, with the IgM isotype being present in similar or smaller proportions than IgG [7,14]. Additionally, both IgG and IgM aPL are not exclusively present in APS, but are also persistently found in patients with other autoimmune rheumatic diseases, such as SLE, and are transiently present during an infection [15,16,17].

Prioritisation of the IgG over the IgM isotype was established due to the higher specificity of the IgG isotype and stronger association with APS manifestations [9]. This redefinition, although intended to improve research consistency, has significant diagnostic implications, as classification criteria are often inappropriately used in clinical practice [18]. In real life, patients with isolated or predominant IgM positivity may be excluded from diagnosis and treatment despite presenting with clinical features of APS. Even though considered, but not sufficient for classification criteria, IgM isotypes form a part in diagnostic and stratification scores such as GAPSS for general APS and EUREKA specific for OAPS patients [19,20]. In GAPSS, IgM isotypes are considered equal to IgG, but in EUREKA, they are less relevant.

Another aspect would be those seronegative patients who are suffering from unequivocal APS clinical manifestations without the persistent presence of aPL above the classification threshold. These patients can test positive for ‘non-criteria’ aPL, such as anti-phosphatidylserine/prothrombin antibodies (aPS/PT), aβ_2_GPI Domain I, anti-annexin A5, or IgA isotypes of aCL and aβ_2_GPI, even in high titres. These ‘non-criteria’ aPL, although not part of the current classification criteria, are increasingly recognised for their diagnostic and stratification value in APS [21,22]. aPS/PT seems to be the most promising. It is hypothesised to be a valuable risk factor of APS pathology, allowing improved patient stratification due to it is high association with APS manifestations, as well as a useful tool to approximate LA positivity [23,24,25,26].

In this scoping review, we synthesise evidence from the recent literature on the prevalence, associations, and clinical value of IgM isotypes of aCL, aβ_2_GPI, and aPS/PT antibodies across various APS phenotypes, including obstetric and thrombotic subsets, paediatric cases, and non-criteria manifestations. We also explore and discuss the current state of APS research and its limitations.

## 2. Methodology

We followed PRISMA-ScR reporting guidance as described in Tricco et al. [27]. The PRISMA-ScR Checklist is provided as Appendix A. Methodological shortcuts inherent to rapid reviews are detailed below. No review protocol was registered because of the accelerated timeline.

### 2.1. Eligibility Criteria

Original human studies with ≥15 participants that reported IgM antiphospholipid antibody (aPL) prevalence or associations with APS manifestations were eligible. We prioritised primary APS (PAPS) cohorts but included mixed APS and non-APS cohorts if IgM data were extractable. Exclusions: case reports, reviews, guidelines, in vitro or animal studies, non-English publications, and papers focused solely on assay techniques or non-IgM isotypes.

### 2.2. Information Sources and Search Strategy

PubMed was searched twice (16 June 2025, and 19 June 2025) using combinations of “APS”, “IgM”, “β2-glycoprotein I”, “phosphatidylserine/prothrombin” and “anticardiolipin”. The complete search strings are provided in Appendix A. Other databases were not accessible within the project timelines. Citation chaining of key references identified nine additional records on 25 June 2025. The final update was performed on 26 June 2025.

### 2.3. Selection Process

Figure 2 shows the PRISMA-ScR flow. Of the 398 database records, three duplicates were removed, resulting in 395 titles/abstracts screened. As the searches were performed on different days, they were not pooled together. When reviewing, duplicates were removed only if encountered by the same reviewer. Ninety-nine records were excluded (case reports = 58, reviews = 21, non-English = 20). Full text of 297 articles was assessed; 257 failed the eligibility criteria shown in Figure 1. For example, excluded studies were those that (i) included aPL data, but the cohort was made of patients with other diseases, such as SLE, (ii) had no IgM data or no IgM data separate from IgG, (iii) focused on comparing different detection strategies rather than APS diagnosis or pathology, (iv) included other aPL than those studied, or (v) other reasons. The “Other” category pooled studies with <15 participants, in vitro studies, or guidelines. As a result, 40 studies were included.

Titles and abstracts were screened by three reviewers (M.O., A.A-L., C.A.); a 10% random sample was double-checked by a senior reviewer (F.M-M.) with 90% concordance. Full texts were assessed by M.O., with 10% rechecked (94% concordance). Because a prospective screening log was not maintained, the search was re-exported on 14 July 2025, and each record was retrospectively assigned to exclusion categories.

### 2.4. Data Charting

Data were extracted into a pre-piloted spreadsheet, capturing publication year, study type (prospective/retrospective), cohort size, cohort characteristics, APS subtype (obstetric/thrombotic/mixed), assay platform, cut-off, IgM prevalence, and reported clinical associations, conclusions, and limitations. Extraction was performed by one reviewer (M.O.) and verified on a 15% sample by F.M-M and A.A-L.

### 2.5. Synthesis of Results

Firstly, for the prevalence data, findings were tabulated; no quantitative pooling or formal risk-of-bias appraisal was undertaken due to assay heterogeneity and the scope of the rapid review. Secondly, for the clinical associations data, a forest plot depicting odds ratios (ORs) was created, including the OR and 95% confidence intervals. In both cases, the values were stratified by APS clinical manifestations.

### 2.6. Methodological Shortcuts and Implications

Due to the rapid scoping review nature, several pragmatic shortcuts were necessary. The search was limited to PubMed, and title and abstract screening were performed by a single reviewer, with a 10% random sample cross-checked by a senior author. Exclusion reasons were assigned retrospectively after re-exporting the search results, and the heterogeneity of the assays disallowed both a formal risk-of-bias appraisal and a meta-analysis.

## 3. Results

### 3.1. Prevalence of aPL IgM Isotype Across Clinical Manifestations

aPL profiles across patients are quite varied. Initially, we examined the prevalence of aCL, aβ_2_GPI, and aPS/PT antibodies across various clinical manifestations to identify intra-manifestation trends and to determine if there is a specific condition where the IgM isotype stands out. In addition, we aimed to explore the relationships between different aPL prevalence rates and whether a particular aPL pattern is associated with any specific clinical manifestation. We focused on cohorts that differentiated between aPL isotypes. The results are grouped by manifestations and summarised below in Table 1, Table 2, Table 3, Table 4 and Table 5. The data are represented as a percentage prevalence of the studied cohort. To provide additional context for the study design, we include information on the sample size, analytical methodology, and cut-offs used.

#### 3.1.1. Obstetric Manifestations

Across these investigations, IgM-class aPL are consistently detected (Table 1). Reported aCL-IgM prevalences range from 3.6% to 81.5%, with six of the nine studies documenting IgM frequencies that equal or surpass their IgG counterparts. The most pronounced disparity is seen in low cut-off testing that yielded 80.4%/81.5% IgM vs. 10.9%/9.8% IgG [28,29]. aCL IgM remains dominant over aCL IgG, even in the medium-high titre group with aPL levels > 40 units [29,30] and the 99th percentile [31,32], underscoring a biological rather than analytical origin. aβ_2_GPI-IgM was similarly frequent, spanning 2–63%. One study, using a 99th percentile cut-off, demonstrated 63% IgM with only 2% IgG [31], whereas others reported substantial levels of both isotypes [30,32]. aPS/PT-IgM prevalence was reported with only one study reaching 7.1%, indicating that it is present, but under-studied and/or under-reported in purely obstetric cohorts. LA positivity varied from 3% to 62% without a discernible relationship to IgM prevalence, suggesting a largely independent prevalence relationship.
jcm-14-07164-t001_Table 1Table 1Prevalence of aPL across obstetric manifestations, expressed as a percentage. ^†^ = at least two positive readings more than 12 weeks apart. ^‡^ = single positivity. ^#^ = single positivity for aPL, but not isotype-differentiated. ^§^ = not all patients were presented for aPL detection (for the actual numbers of patients used per aPL, visit the original publication), aβ_2_GPI: anti-β_2_-glycoprotein I; aCL: anti-cardiolipin; aPL: antiphospholipid antibodies, APS: antiphospholipid syndrome; aPS/PT: anti-phosphatidylserine/prothrombin; AU: arbitrary units; ELISA: enzyme-linked immunosorbent assay; GPL: IgG phospholipid level; HD: healthy donor; LA: lupus anticoagulant; MPL: IgM phospholipid level; n or N = cohort size; NA = not available; OAPS: obstetric APS; PAPS: primary APS; +: positive.Prevalence of LA, aCL, aβ_2_GPI, and aPS/PT Across Obstetric Manifestations [%]ReferenceYearCohortAssay TypePersistency ^†^Cut-OffNLAaCLaβ_2_GPIaPS/PTIgGIgMIgGIgMIgGIgMChen et al. [28]2024aPL+ patients with obstetric complicationsNANA≥20–40 GPL/MPL/AU923.310.980.43.314.1

Chen et al. [29]2024118 aPL+ patients, with 124 pregnanciesNANALow: ≥20–40 GPL/MPL/AU923.89.881.53.314.1

Medium–high: ≥40–80 GPL/MPL/AU327.421.937.540.631.3

Long et al. [33]2023OAPS patients (nPAPS = 123)Commercial ELISAYes40 AU20950.235.911.034.024.4

Liu et al. [31]2022APS patientsCommercial ELISAYes99th percentile240 ^‡^15.06.713.32.162.9

Alijotas-Reig et al. [34]2019OAPS patientsCommercial/In-house ELISAYes99th percentile706 ^#^50.421.115.311.58.9

Žigon et al. [35]2015Patients with OAPS clinical manifestationsIn-house ELISANA99th percentile (222 HD)169 ^§^8.710.13.65.92.49.57.1Pereira et al. [32]2015OAPS patientsIn-house ELISAYes99th percentile7116.939.449.729.623.9

Bouvier et al. [30]2014OAPS patientsIn-house ELISAYes>42.1 GPL/MPL/AU51761.747.271.922.140.6

Roye-Green et al. [36]2011Women with a history of 2 or more consecutive spontaneous abortionsIn-house ELISANAaCL (Low): 10–20 GPL/MPLaβ_2_GPI: >20 AU50221062

aCL (Medium): 21–80 GPL/MPL216


#### 3.1.2. Combined Thrombotic and Obstetric Manifestations

In the eleven studies of thrombotic and obstetric manifestations (Table 2), reported aCL-IgM prevalences range from 4.8% to 40.1%, uniformly lower than the parallel aCL-IgG figures (11.9–71.6%). aβ_2_GPI-IgM appears in 3.9–44.7% of patients and is lower than aβ_2_GPI-IgG in every study except in one study, where IgM marginally exceeds IgG (28% vs. 25%) [37]. aPS/PT-IgM, documented in eight studies, exhibits the highest frequencies—ranging from 31.1% to 79.1%—and often surpasses its IgG counterpart.

LA positivity ranges from 32.9% to 97.3%. A cohort with very high LA levels presented the lowest aCL-IgM and aβ_2_GPI-IgM frequencies (4.8%, 3.9%, respectively) [38], whereas those with intermediate LA proportions (approx. 33–60%) report aCL-IgM and aβ_2_GPI-IgM to be 18.3–40.1% and 23.9–44.7%, respectively.

We observed no concordance in the prevalence of IgM positivity between studies with the same cut-offs; the IgM prevalences were highly variable, regardless. Collectively, these data indicate that IgM-class antibodies—particularly aPS/PT-IgM—are prevalent in mixed thrombotic–obstetric APS, and their reported frequencies depend on the studied cohort, assay type, and cut-off definition.
jcm-14-07164-t002_Table 2Table 2Prevalence of aPL across combined obstetric and thrombotic manifestations, expressed as a percentage. ^†^ = at least two positive readings more than 12 weeks apart. ^#^ = single positivity for aPL, but not isotype differentiated. ^§^ = not all patients were presented for aPL detection (for the actual numbers of patients used per aPL, visit the original publication), aβ_2_GPI: anti-β_2_-glycoprotein I; aCL: anti-cardiolipin; APS: antiphospholipid syndrome; aPS/PT: anti-phosphatidylserine/prothrombin; AU: arbitrary units; CLIA: chemiluminescent immunosorbent assay; ELISA: enzyme-linked immunosorbent assay; GPL: IgG phospholipid level; HD: healthy donor; LA: lupus anticoagulant; LIA: line immune assay; MPL: IgM phospholipid level; N or n = cohort size; PAPS: primary APS; SAPS: secondary APS; SLE: systemic lupus erythematosus.Prevalence of LA, aCL, aβ_2_GPI and aPS/PT Across Combined Thrombotic and Obstetric Manifestations [%]ReferenceYearCohortAssay TypePersistency ^†^Cut-OffNLAaCLaβ_2_GPIaPS/PTIgGIgMIgGIgMIgGIgMZhao et al. [38]2024APS patients meeting ACR/EULAR criteriaCommercial ELISAYesMedium-high: 40–79 GPL/MPL/AU33597.311.94.88.13.9

High: 80 GPL/MPL/AU28.14.823.03.9

Egri et al. [39]2021APS patients (nPAPS = 17, nSAPS = 8)aCL, aβ_2_GPI: CLIA aPS/PT: Commercial ELISAYesaCL, aβ_2_GPI: 20 GPL/MPL/AU aPS/PT: 30 AU25 ^§^58642576322872Liu et al. [40]2020APS patients (nPAPS = 88, nSAPS = 104)Commercial ELISAYesaPS/PT: 30 AU19269



7173Volkov et al. [41]2020APS patients (nPAPS = 63, nSAPS = 67)LIA stripYes95th percentile1308459266632

Heikal et al. [37]2019APS patients (nPAPS = 51, nSAPS with associated SLE = 20)Commercial ELISAYesaCL, aβ_2_GPI: 20 GPL/MPL/AU71 ^§^71253425283142aPS/PT: 30 AUShi et al. [42]2018APS patientsaPS/PT: Commercial ELISAYesaPS/PT: 30 AU67 (PAPS)




70.179.1119 (SAPS)73.160.5Hoxha et al. [43]2017PAPS patientsaCL, aβ_2_GPI: In-house ELISAYesaCL, aβ_2_GPI: 99th Percentile (100 HD)19757.769.040.171.644.729.948.2aPS/PT: Commercial ELISAaPS/PT IgG/IgM: 61.4 AU/56.3 AUShi et al. [44]2017APS patientsCommercial ELISAYesaCL: 40 GPL/MPLaβ_2_GPI: 99th Percentile252 ^#^32.956.325.037.327.8

Hoxha et al. [25]2015PAPS patientsaCL, aβ_2_GPI: In-house ELISAYesaCL, aβ_2_GPI: 99th PercentileaPS/PT: 30 AU16046.866.338.167.544.43047.5aPS/PT: Commercial ELISAVlagea et al. [45]2013APS patientsIn-house ELISAYesaPS/PT IgG/IgM: 10 AU/15 AU95 (PAPS)




35.732.645 (SAPS)40.031.1Boffa et al. [46]2009APS patientsIn-house and commercial ELISAYes99th Percentile10937.571.618.333.023.9




#### 3.1.3. Purely Thrombotic Manifestations

Six independent studies discussed the prevalence of aPL in thrombotic manifestations (Table 3). Depending on the cut-off used, the reported aCL-IgM prevalences range from 1.75% to 70.5%, consistently lower than the corresponding aCL-IgG values. aβ_2_GPI-IgM, was found in 0–58.8% cases, with only one study reporting the IgM isotype exceeding IgG (50.9% vs. 45.6%) [47]. Zen et al. compared cohort classification according to Sydney and ACR/EULAR criteria and showed that aCL and aβ_2_GPI IgM isotypes in medium-to-high titres were highly prevalent only in Sydney criteria-classified PAPS (70.5% and 58.8%, respectively), even with low LA prevalence (5.9%) [18]. The IgM prevalence in this cohort was higher than in the ACR/EULAR-classified cohort, where the prevalence of aCL and aβ_2_GPI IgM reached only 42.7% and 35.7%, respectively.

Interestingly, Masson et al. demonstrated that the IgM isotype is more prevalent in PAPS patients diagnosed at or after the age of 65 than in those diagnosed at an earlier age [48]. Although not all patients were tested for each aPL, making the results not entirely comparable, it appears that the prevalence of IgM doubles and the prevalence of IgG halves when patients are diagnosed after the age of 65.

Comparing three purely TAPS cohorts of Uludağ et al. [49], Vandevelde et al. [50], and del Ross et al. [51], the IgM prevalences were similar, most likely due to the comparable sample sizes (105, 197, and 106, respectively) and cut-offs. Prevalences from studies that utilised the same assay method [49,50], commercial enzyme-linked immunosorbent assay (ELISA), were more concordant than the study that utilised an in-house ELISA [51].

Moreover, PS/PT-IgM positivity has been reported in two studies and reached just above half of the study cohorts, with IgM isotype being the equally or slightly more prevalent isotype [49,50]. Studies that utilised in-house ELISA with a 99th percentile cut-off report high IgM prevalence [18,51]. Overall, IgM antibodies—particularly those directed against β_2_GPI and PS/PT—are regularly documented in TAPS, and reported prevalences vary depending on the assay platform and cut-off definition.
jcm-14-07164-t003_Table 3Table 3Prevalence of aPL across thrombotic manifestations, expressed as a percentage. ^†^ = at least two positive readings more than 12 weeks apart. ^§^ = not all patients were presented for aPL detection (for the actual numbers of patients used per aPL, visit the original publication), aβ_2_GPI: anti-β_2_-glycoprotein I; aCL: anti-cardiolipin; aPL: antiphospholipid antibodies, APS: antiphospholipid syndrome; aPS/PT: anti-phosphatidylserine/prothrombin; AU: arbitrary units; ELISA: enzyme-linked immunosorbent assay; GPL: IgG phospholipid level; LA: lupus anticoagulant; MPL: IgM phospholipid level; n or N = cohort size; PAPS: primary APS; SAPS: secondary APS; SLE: systemic lupus erythematosus.Prevalence of LA, aCL, aβ_2_GPI and aPS/PT Across Thrombotic Manifestations [%]ReferenceYearCohortAssay TypePersistency ^†^Cut-OffNLAaCLaβ_2_GPIaPS/PTIgGIgMIgGIgMIgGIgMZen et al. [18]2025PAPS PatientsIn-house ELISAYesaCL IgG/IgM (Medium): 21.1 GPL/27.2 MPL171 (ACR/EULAR PAPS)/34 (only Sydney PAPS)81.3/5.90/20.51.75/5.90.58/17.20/2.9

Medium-to-high: 99th percentile81.3/042.7/70.578.4/035.7/58.8Uludağ et al. [49]2023TAPS Patients Commercial ELISAYesaCL: 40 GPL/MPL aβ_2_GPI: 20 AU aPS/PT: 30 AU10584.860.026.744.830.555.255.2Vandevelde et al. [50]2022TAPS PatientsCommercial ELISAYesaCL, aβ_2_GPI: 20 GPL/MPL/AU19794.943.127.435.026.448.761.9aPS/PT: 30 AUMasson et al. [48]2022APS PatientsELISAYes40 AU or 99th percentile127 (<65 years old) ^§^66.938.518.135.017.5

57 (≥65 years old) ^§^52.816.433.914.331.4Pérez et al. [47]2018APS Patients (nPAPS = 35, nSAPS with SLE = 22)Commercial ELISAYes20 GPL/MPL/AU5766.742.143.945.650.9

Del Ross et al. [51]2015TAPS PatientsIn-house ELISAYes99th percentile10671.777.439.673.640.6




#### 3.1.4. Thrombotic and Other Manifestations

This group of six studies (Table 4) looked at other manifestations (thrombocytopenia [52], cardiac [53], systemic non-criteria manifestations [54], and pulmonary events [55]) and paediatric cohorts with criteria and non-criteria manifestations [56,57]. Across these cohorts, aCL-IgM prevalences range from 23.5% to 64.0% and are consistently lower than the parallel aCL-IgG figures (26–64.9%). aβ_2_GPI-IgM occurs in 20.7–47% of patients and generally trails the corresponding IgG (30.7–64.9%). aPS/PT-IgM was assessed only in the paediatric series, reaching 68% and exceeding aPS/PT-IgG (58%) [56]. LA positivity spans 49–91%, without a discernible relationship to IgM frequencies. Paediatric patients appear to have higher LA prevalence, while the aCL-IgM and aβ_2_GPI-IgM prevalences remain lower.

Methodological choices modulate IgM detection. Cohorts that used low analytical thresholds [54,55,57] report concordant results with studies that used higher cut-offs, with only aCL-IgM showing marginally higher prevalence. There is no clear evidence for a higher cut-off decreasing aPL prevalence. All studies relied on commercial assays. Nevertheless, inter-cohort variability in [52] vs. [53] produced up to a two-fold difference in aCL-IgM detection with almost an exact cut-off. Collectively, these data demonstrate that IgM-class aPL—particularly aCL and aβ_2_GPI, with notable aPS/PT in children—are standard in APS cohorts that combine thrombotic and miscellaneous systemic features, and their reported frequencies are strongly influenced by assay threshold selection.
jcm-14-07164-t004_Table 4Table 4Prevalence of aPL across thrombotic and other manifestations, expressed as a percentage. ^†^ = at least two positive readings more than 12 weeks apart. aβ_2_GPI: anti-β_2_-glycoprotein I; aCL: anti-cardiolipin; aPL: antiphospholipid antibodies, APS: antiphospholipid syndrome; aPS/PT: anti-phosphatidylserine/prothrombin; AU: arbitrary units; ECLIA: electrochemiluminescence immunoassay; ELISA: enzyme-linked immunosorbent assay; GPL: IgG phospholipid level; LA: lupus anticoagulant; MPL: IgM phospholipid level; N or n = cohort size; PAPS: primary APS; SAPS: secondary APS; TP: thrombocytopenia; +: positive; −: negative.Prevalence of LA, aCL, aβ_2_GPI and aPS/PT Across Thrombotic and Other Manifestations [%]ReferenceYearCohortAssay TypePersistency ^†^Cut-OffNLAaCLaβ_2_GPIaPS/PTIgGIgMIgGIgMIgGIgMSloan et al. [56]2024Paediatric APS patients (nPAPS = 11, nSAPS = 8)Commercial ELISAYesaCL, aβ_2_GPI: 20 GPL/MPL/AUaPS/PT: 30 AU198437264247586840 GPL/MPL/AU261626265868Shi et al. [52]2023PAPS patientsCommercial ELISAYes40 GPL/MPL/AU74 (TP +)90.564.927.064.940.5

144 (TP −)67.440.323.649.340.3Djokovic et al. [53]2022PAPS patientsCommercial ELISAYes41 GPL/MPL/AU36055.029.747.830.340.8

Morán-Álvarez et al. [57]2022aPL+ paediatric patientsELISA, ECLIAYesaCL: 18 GPL/MPLaβ_2_GPI: 10 AU8277.830.523.531.720.7

Stojanovich et al. [54]2013APS patientsCommercial ELISAYes11 GPL/MPL/AU260 (PAPS)51.236.554.231.937.7

114 (SAPS)49.159.664.043.044.7Stojanovich et al. [55]2012APS patientsCommercial ELISANo, only 6 weeks10 GPL/MPL/AU214 (PAPS)60.228.050.533.631.8

115 (SAPS)48.240.049.641.739.1


#### 3.1.5. Cerebrovascular Manifestations

Three studies evaluated IgM aPL in cerebrovascular APS (Table 5). In an isolated aPS/PT stroke cohort, aPS/PT-IgM was detected in 85% of LA-negative versus in only 25% of LA-positive patients [58]. A second series distinguished two subgroups: an unselected APS cohort (aCL-IgM 24.3%, aβ_2_GPI-IgM 26.4%) and an isolated IgM subset with markedly higher prevalences (aCL-IgM 91.7%, aβ_2_GPI-IgM 62.5%) [59]. In a third cohort, comparing PAPS alone with PAPS plus rheumatic fever, aCL-IgM prevalence was similar across groups, whereas aβ_2_GPI-IgM was absent [60].
jcm-14-07164-t005_Table 5Table 5Prevalence of aPL across cerebrovascular manifestations, expressed as a percentage. ^†^ = at least two positive readings more than 12 weeks apart. aβ_2_GPI: anti-β_2_-glycoprotein I; aCL: anti-cardiolipin; APS: antiphospholipid syndrome; aPS/PT: anti-phosphatidylserine/prothrombin; AU: arbitrary units; ELISA: enzyme-linked immunosorbent assay; GPL: IgG phospholipid level; LA: lupus anticoagulant; MPL: IgM phospholipid level; N or n = cohort size; PAPS: primary APS; RF: rheumatic fever; SLE: systemic lupus erythematosus; +: positive; −: negative.Prevalence of LA, aCL, aβ_2_GPI and aPS/PT Across Cerebrovascular Manifestations [%]ReferenceYearCohortAssay TypePersistency ^†^Cut-OffNLAaCLaβ_2_GPIaPS/PTIgGIgMIgGIgMIgGIgMRadin et al. [58]2021Patients with a suspicion for APS or SLE negative for aCL and aβ_2_GPICommercial ELISAYes40 AU22 (LA +)100000036.425.020 (LA −)090.485.0Urbanski et al. [59]2018APS patientsCommercial ELISAYes12 GPL/MPL/AU14478.580.624.353.526.4

24 (Isolated IgM)0091.7062.5Camargo et al. [60]2012APS patientsIn-house ELISAYes40 GPL/MPL/AU5 (APS-RF)404060600

68 (PAPS)7938532510


### 3.2. Associations of IgM aPL with Clinical Manifestations

Prevalence does not necessarily signal clinical value, so we extracted the probabilities of outcomes for different aPL from studies where this information was available. The dataset presents a detailed collection of odds ratios (ORs) and their associated confidence intervals (CIs), assessing the association between various IgM aPL and specific APS-related clinical manifestations. The values span a broad range of pathological contexts—obstetric, thrombotic, and mixed—providing insights into the diagnostic and potentially pathogenic relevance of these antibodies. The N value associated with each manifestation represents the number of patients with the given clinical manifestation. Only in the cases of isolated IgM aPL results, the N is manifestation- and aPL-specific.

#### 3.2.1. Obstetric Manifestations

##### Significant Associations of IgM aPL

As shown in Figure 3, among 43 study-level contrasts, 15 reached statistical significance of the association of aPL IgM isotype positivity with any poor obstetric outcome. For pregnancy loss, increased odds were reported with IgM aCL (OR 4.26, 95% CI 1.70–10.75), aβ_2_GPI (OR 4.87, 2.48–9.60), and aPS/PT (OR 4.71, 2.79–8.12) [42], whereas other cohorts documented decreased odds with aPS/PT (OR 0.60, 0.50–0.70) [61] and with aβ_2_GPI (OR 0.70, 0.60–0.70) [35]. Embryonic loss was positively associated with aCL (OR 2.66, 1.06–6.71) [62] and inversely associated with aβ_2_GPI (OR 0.55, 0.30–0.89) [30]. Among women with obstetric complications and suspected APS, inverse associations were also seen for aβ_2_GPI with ≥3 consecutive miscarriages at <10 weeks of gestation (WG) (OR 0.40, 0.40–0.50) and with foetal death at >10 WG (OR 0.50, 0.40–0.60), and for aCL with premature birth at <34 WG (OR 0.70, 0.70–0.80) [35]. Positive aCL associations were reported for placenta-mediated complications (OR 2.02, 1.09–3.73), placental abruption (OR 5.53, 1.24–24.55), and small-for-gestational-age neonate (OR 2.02, 1.06–3.85) [30,62]. For preeclampsia (PE), directionality differed by case definition: lower odds with aCL in early-onset PE (OR 0.30, 0.10–0.80) [63], but higher odds when onset was not specified (OR 5.06, 1.15–22.30) [62].

##### Lack of Significant Associations of IgM aPL

A series of studies reported insignificant associations of at least one IgM aCL, aβ_2_GPI, or aPS/PT positivity with pregnancy loss [35,45,61], embryonic loss [62], ≥three consecutive miscarriages at <10 WG [35,64], foetal death at >10 WG [35,64], premature birth <34 WG [35,64], early-onset PE [30,63,64], PE [30,62], foetal growth restriction [62,64], and SGA neonate ([30] for aβ_2_GPI). Although Žigon et al. [35] reported an elevated OR for IgM aPS/PT in pregnancy loss (OR 9.00), with the lower limit including zero, it had to be considered as not significant.

#### 3.2.2. Thrombotic and Vascular Manifestations

##### Significant Associations of IgM aPL

Across the 43 study-level contrasts extracted from the thrombosis dataset, 16 showed statistical significance of aPL IgM positivity associated with thrombotic or vascular events (Figure 4). For general thrombosis, significant positive associations were reported for aPS/PT at both >43 AU (OR 2.81, 95% CI 1.83–4.32) and >200 AU (OR 6.48, 3.28–12.80) [50]; for aCL (OR 3.93, 1.61–10.55), aβ_2_GPI (OR 3.22, 1.72–6.24) and aPS/PT (OR 4.24, 2.78–6.56) [42]; for aPS/PT (OR 3.80, 1.90–7.70) [43]; and for aPS/PT in LA-negative sera (OR 5.40; 1.80–16.10) [65]. Only a single study reported a positive association for arterial thrombosis with aPS/PT (OR 3.12, 1.89–5.23) [42]. For venous thrombosis, increased odds were observed with aCL (OR 3.27, 1.32–7.92), aβ_2_GPI (OR 2.68; 1.38–5.11), and aPS/PT (OR 3.24, 2.02–5.29) [42], with the positive association of aPS/PT (OR 2.54, 1.35–4.77) being confirmed in another study [45]. For maternal thromboembolism, lower odds were reported with aCL (OR 0.20, 0.04–0.90) [63]. In microangiopathy, aPS/PT was associated with higher odds (OR 3.60; 1.80–7.40) [43].

Markedly increased odds were noted for the isolated IgM aCL/aβ_2_GPI phenotype. Firstly, for retinal thrombosis (OR 27.60, 2.61–291.10) [51], and secondly for stroke (OR 3.80, 1.30–11.50) [59].

##### Lack of Significant Associations of IgM aPL

Simultaneously, a series of studies did not report a statistically significant association of IgM aPL positivity with thrombotic or vascular events. For example, in thrombosis, a non-significant IgM association were noted by [51,61,66] and the LA-positive subgroup of [65]. For arterial thrombosis, non-significant findings were reported for aCL, aβ_2_GPI, and aPS/PT in [61]; for aCL and aβ_2_GPI in [66]; for aCL and aβ_2_GPI in [42]; and for isolated IgM aCL/aβ_2_GPI in [51]. Association with venous thrombosis was non-significant for all aPL in [61]; for aCL and aβ_2_GPI in [66]; and for isolated IgM aCL/aβ_2_GPI in [51]. In addition, isolated IgM aCL/aβ_2_GPI showed no significant association with cerebrovascular disease, pulmonary embolism, or retinal thrombosis [51]. Finally, microangiopathy, thrombocytopenia, and stroke were non-significant across all three specificities in [61].

### 3.3. New Approach to aPL Associations: Cluster Analysis

Apart from the OR analysis, Long et al. employed a different method to evaluate the association of aPL with obstetric manifestations [33]. They applied hierarchical clustering to a cohort of 209 persistently aPL-positive patients, integrating demographic data, clinical manifestations, obstetric history, and laboratory results to uncover distinct APS phenotypes. They identified four discrete clusters; the largest, Cluster 3 (n = 87), was characterised by an overwhelming predominance of IgM isotypes—83.9% were positive for aβ_2_GPI-IgM and 33.3% for aCL-IgM. Clinically, Cluster 3 patients experienced the highest rate of early miscarriage (<10 WG) at 60.9%, significantly exceeding that of other clusters, while experiencing the lowest incidence of late miscarriage (>10 WG) at 31.0%. Unlike other clusters, these women exhibited minimal thrombotic complications—arterial, venous, or microvascular—and carried the lowest cardiovascular risk burden. Only 8% had concomitant SLE, and a mere 7% tested positive for LA. Placental histopathology of Cluster 3 further distinguished as the placentas displayed suggestive signs of subclinical microvascular injury.

### 3.4. Factors Contributing to Thrombosis: Neutrophil Activation and NETosis

Thrombosis in APS patients can be initiated and promoted in multiple ways. One of them is through a specialised form of neutrophil death called NETosis, where activated neutrophils release their decondensed chromatin into the extracellular space, forming neutrophil extracellular traps (NETs) [67]. Although most investigations of NETs focus on IgG aPL, Bouvier et al. in 2024 found no difference in a surrogate NETosis marker (NEUT-RI score) between aPL-positive and aPL-negative patients [68]. The Neutrophil-Reactive Intensity (NEUT-RI) score measures the metabolic activity of neutrophils, which is typically higher in activated neutrophils, as they produce cytokines and other inflammatory mediators [69]. Intriguingly, patients with high-titre IgM aCL or aβ_2_GPI had lower NEUT-RI values (OR (aCL IgM) = 0.86 [0.77–0.97], *p* = 0.0097) and (OR (aβ_2_GPI-IgM) = 0.73 [0.63–0.84], *p* = 0.0015), hinting at a possible protective effect of some IgM isotypes against neutrophil activation. On the other hand, Sloan et al. used a distinct NETosis marker, calprotectin levels, which significantly positively correlated with aPS/PT IgM levels in paediatric APS patients [56].

## 4. Discussion

This scoping review aimed to map and critically synthesise peer-reviewed evidence published between Jan 2000 and June 2025 on the prevalence, clinical associations, and potential diagnostic value of IgM-class aCL, aβ_2_GPI, and aPS/PT antibodies. IgM antibodies were consistently detected across APS phenotypes, yet their clinical significance remains controversial, particularly after their exclusion as stand-alone criteria in the 2023 ACR/EULAR classification. Important knowledge gaps remain, including the scarcity of well-characterised isolated IgM cohorts, heterogeneity of assays and thresholds, and the lack of prospective outcome data. While our review is limited by its PubMed-only, English-language scope and the absence of formal risk-of-bias grading, it nonetheless provides the most comprehensive synthesis to date for clinicians considering how to interpret IgM positivity in APS.

In obstetric cohorts, IgM prevalence often equals or exceeds IgG reaching up to 82% for aCL, while thrombotic and mixed cohorts typically show rates of 20–45%. aPS/PT-IgM remains the least studied yet reaches up to 80% in some mixed and paediatric series. Importantly, high IgM frequencies persist under both liberal and stringent assay thresholds, suggesting a true biological signal rather than an analytical artefact.

High prevalence alone does not imply pathogenicity; therefore, the odds-ratio analysis provides essential context of whether aPL are or are not significantly associated with a given manifestation. In our review, positive associations appeared across several obstetric endpoints, with aCL-IgM showing the most frequent positive signals across pregnancy loss and placenta-mediated complications. Some complications appeared as significantly not-associated with aβ_2_GPI-IgM (OR < 1). We interpret these inverse findings as cohort-specific artefacts, since in one key study, only a small fraction of women with obstetric complications were aPL-positive, reducing the power to detect true effects [35]. When viewed in a broader context, other studies echo this heterogeneity: some show platform-robust associations for aCL-IgM with obstetric morbidity [70,71], whereas others report no association [72,73,74]. None of these studies considered an isolated IgM phenotype.

In thrombotic and vascular manifestations, the most informative data come from three studies of isolated IgM serology. aPS/PT-IgM showed a positive association with thrombosis [65], while isolated aCL/aβ_2_GPI-IgM was linked to retinal occlusion [51] and, in a separate cohort, to stroke [59], although sample sizes were small. Outside these isolated settings, most significant associations came from only a handful of studies [42,43,50], raising concerns about generalisability and potential publication bias. Overall, thrombotic risk linked to IgM appears sporadic and context-dependent, with aPS/PT-IgM emerging as the most consistent signal. Independent reviews have likewise concluded that IgM aPL are associated with thrombosis only in a minority of cases, insufficient to justify their use as standalone diagnostic or prognostic markers [75].

Evidence beyond prevalence and association studies suggests that IgM aPL may exert phenotype-specific effects. Hierarchical clustering identified an IgM-predominant obstetric subgroup, characterised by high rates of early miscarriage but minimal thrombotic complications, low cardiovascular risk, and scarce LA positivity [33]. Functional studies provide additional nuance. In one cohort, high-titre aCL- and aβ_2_GPI-IgM correlated with lower neutrophil activation, suggesting a potential protective effect against NETosis [68], whereas aPS/PT-IgM tracked with higher calprotectin in paediatric APS [56]. Together, these findings hint that different IgM specificities may modulate, rather than uniformly amplify, inflammatory and thrombotic pathways.

As with the adoption of new APS classification criteria, some patients previously included will no longer be classified, which rationally raised concerns on patient impact; therefore, many studies investigated the effects of this change [18,76,77,78,79,80,81]. The primary reason for exclusion in many cohorts was isolated positivity for IgM, even in moderate-to-high titres [18,77,78,80]. Especially those with double positivity are those patients necessitating meticulous care due to the high risk of thrombosis [77]. On the other hand, in the cohort of Vasi et al., only 6% of patients were excluded based on isolated IgM positivity [81].

The persistence of poor obstetric outcomes among low-titre IgM carriers underscores the insufficiency of the new ACR/EULAR guidelines that de-emphasise the role of IgM in APS pathology. In practice, a woman with recurrent early miscarriage and isolated low-titre IgM positivity may fail to meet entry criteria for prophylactic anticoagulation, yet her risk profile warrants intervention. Supporting this, prospective analyses have shown that close clinical follow-up is protective against adverse pregnancy outcomes, and that APS diagnosis itself is correlated with improved live birth rates [63,82]. Furthermore, in TAPS, Zen et al. further showed that nearly one-fifth of well-defined PAPS patients would be declassified under the new criteria, with one-third of the excluded isolated IgM group suffering recurrent thrombosis, sometimes despite prophylaxis [18]. These findings indicate that while IgM positivity is often overlooked, it can carry meaningful risk in specific contexts. Wholesale dismissal of IgM, therefore, risks diagnostic blind spots and premature discontinuation of prophylaxis in vulnerable subgroups.

To clarify the pathological role of IgM aPL, patients with isolated IgM positivity must be studied. However, such cases are rare and represent only a small fraction of APS cohorts. Reported prevalences vary, with isolated IgM phenotypes detected in 5.7–12.3% of obstetric patients, 3.5–5.4% of thrombotic patients [70], and up to 12.3–16.7% in other series [34,51,59]. Interpretation is further hampered because many studies pool IgM with other isotypes, while those that do stratify rarely provide detailed per-patient data linking isolated IgM to clinical outcomes. As a result, the independent contribution of IgM remains difficult to disentangle from that of coexisting aPL.

Current APS management is targeting symptomatology and is not directed against aPL isotypes. Long-term anticoagulation for TAPS and low-dose aspirin with or without heparin for OAPS remain the standards of care, with none of them specifically targeting IgM aPL [11]. Several agents may modulate aPL biology or downstream pathways, but evidence is indirect and not IgM-selective. Heparin interferes with aPL-mediated trophoblast and coagulation–surface interactions in experimental systems and improves pregnancy outcomes clinically, yet this is a mechanistic rather than isotype-specific effect [83,84]. In a recent study, heparin could lower aβ_2_GPI-IgM levels while keeping aCL-IgM levels unaffected, but the molecular mechanism remains unknown [85].

This rapid scoping review necessarily privileges breadth over absolute precision. Our single-database, English-only search may have missed regional or non-indexed studies, and the absence of a registered protocol or duplicate screening increases the possibility of selection drift. Exclusion categories were reconstructed post hoc, carrying a small risk of misclassification, while assay heterogeneity rendered both risk-of-bias grading and meta-analysis scientifically unsound. These pragmatic compromises mean that the prevalence bands and odds ratios we report should be read as directional signals rather than final estimates, pending harmonised, prospectively designed investigations.

## 5. Conclusions

The role of IgM aPL—particularly aCL, aβ_2_GPI, and aPS/PT —remains incompletely defined. Given the high seroprevalence, IgM results can contextualise borderline findings, flag mixed-type APS, and inform research phenotyping. IgM-dominant obstetric cluster calls for tailored follow-up. Women with isolated IgM and early miscarriages often lack conventional thrombotic risk factors or LA; they may benefit from closer obstetric surveillance even if not meeting full APS criteria. High-titre aPS/PT-IgM merits attention in thrombotic APS. Significant associations and dose–response relationships suggest added prognostic value, particularly when LA is absent. Accordingly, we advocate retaining IgM measurement in routine aPL panels.

## 6. Future Perspectives

To accurately determine the actual effect of IgM antibodies, prospective, standardised studies are essential. Ideally, incorporating uniform cut-offs, harmonised aPL data presentation stratified by isotype, including specific individual patient descriptions, and mechanistic endpoints such as NETosis, complement activation and placental histology. Future studies should prospectively track IgM-only cohorts to determine the clinical background and outcomes of this rare population. Only with this level of methodological rigour can we determine whether IgM antibodies are innocuous bystanders, phenotype-specific modifiers, or proper pathogenic drivers.

## Figures and Tables

**Figure 1 jcm-14-07164-f001:**
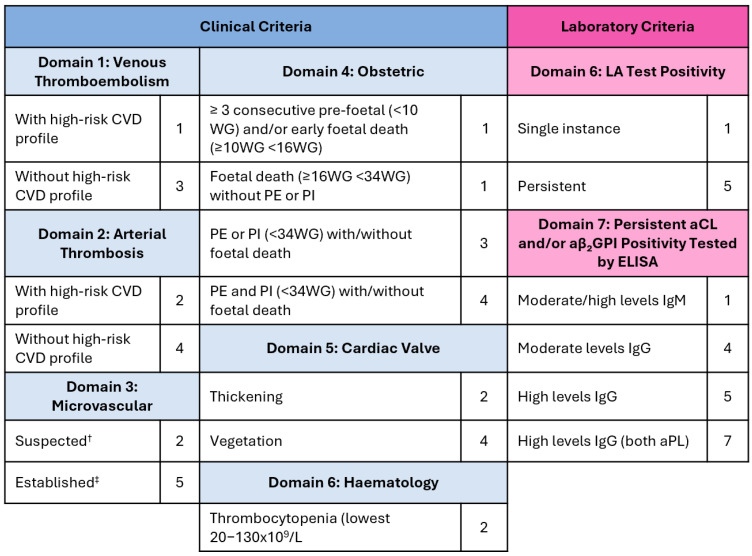
ACR/EULAR criteria for classification of APS. Adapted from [9]. ^†^ Livedo racemosa (by physical examination), Livedoid vasculopathy lesions (by physical examination), antiphospholipid antibody (aPL) nephropathy (by physical examination or laboratory tests), pulmonary haemorrhage (by clinical symptoms and imaging), ^‡^ Livedoid vasculopathy (by pathology), aPL nephropathy (by pathology), pulmonary haemorrhage (by bronchoalveolar lavage or pathology, myocardial disease (by imaging or pathology), adrenal haemorrhage or microthrombosis (by imaging or pathology), Moderate levels equal to 40–79 units, high levels equal to levels ≥ 80 units, aβ_2_GPI: anti-β_2_-glycoprotein I; aCL: anti-cardiolipin; aPL: antiphospholipid antibodies; CVD: cardiovascular disease; ELISA: enzyme-linked immunosorbent assay; LA: lupus anticoagulant; PI: placental insufficiency with severe features, PE: preeclampsia with severe features; WG: weeks of gestation.

**Figure 2 jcm-14-07164-f002:**
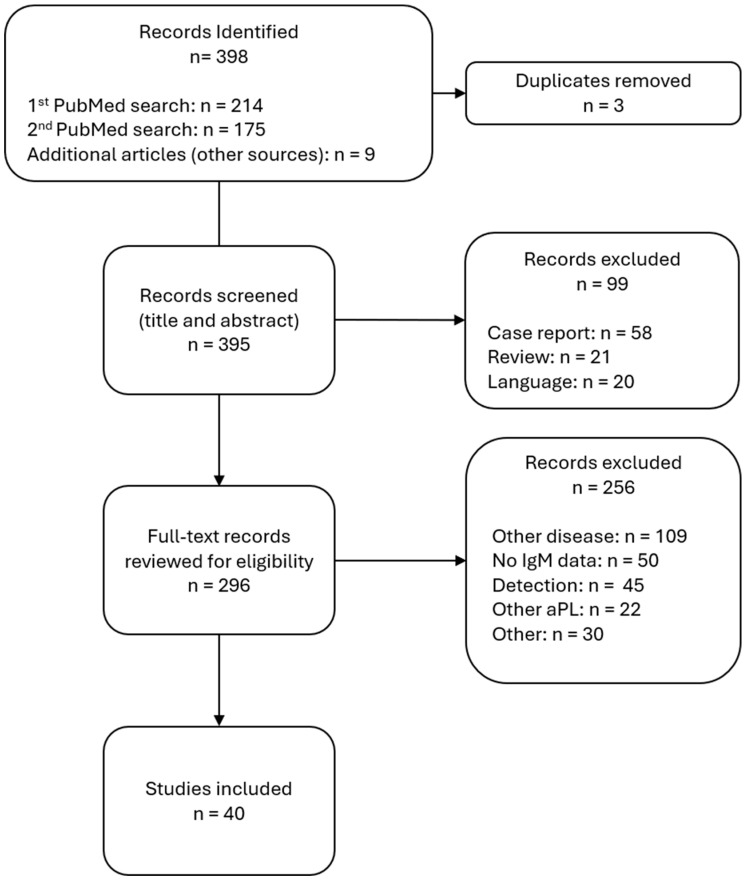
PRISMA-ScR flow diagram of study identification and selection of studies to be included in the review.

**Figure 3 jcm-14-07164-f003:**
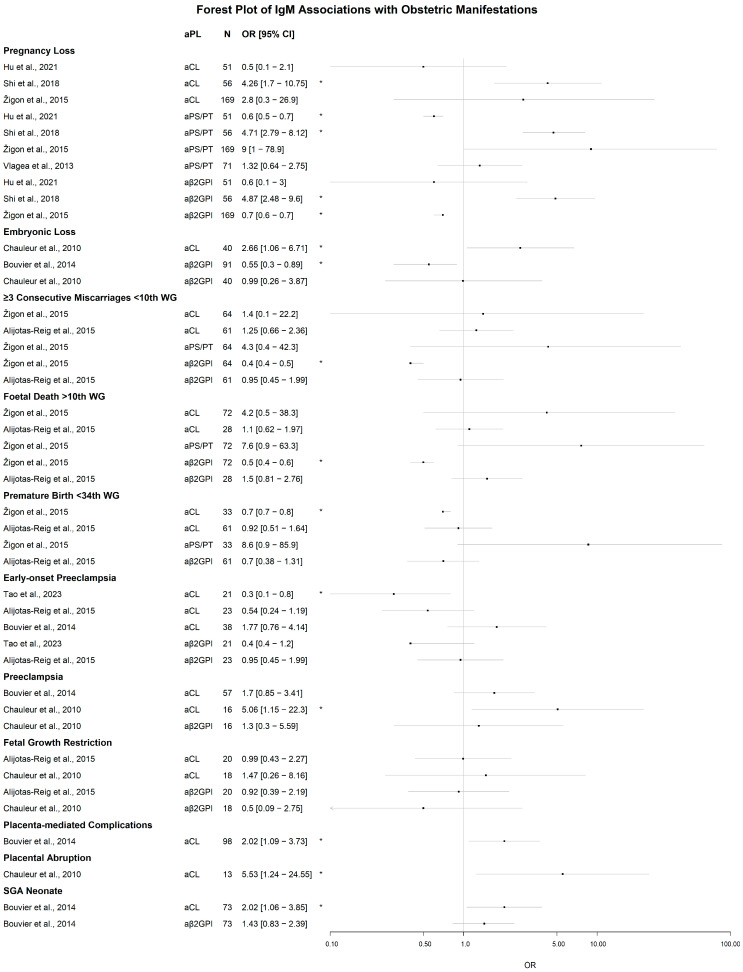
Forest plot of the odds ratios (ORs) of IgM aPL in different obstetric clinical manifestations. Data are ordered by aPL and represented as OR [95% confidence interval]. Significant results are marked with *. N represents the number of patients with the particular clinical manifestation. The scale is logarithmic. aβ_2_GPI: anti-β_2_-glycoprotein I; aCL: anti-cardiolipin; aPS/PT: anti-phosphatidylserine/prothrombin; SGA: small for gestational age; WG: week of gestation. Cited works: Hu et al. [61], Žigon et al. [35], Vlagea et al. [45], Bouvier et al. [30], Shi et al. [42], Chauleur et al. [62], Alijotas-Reig et al. [64] and Tao et al. [63].

**Figure 4 jcm-14-07164-f004:**
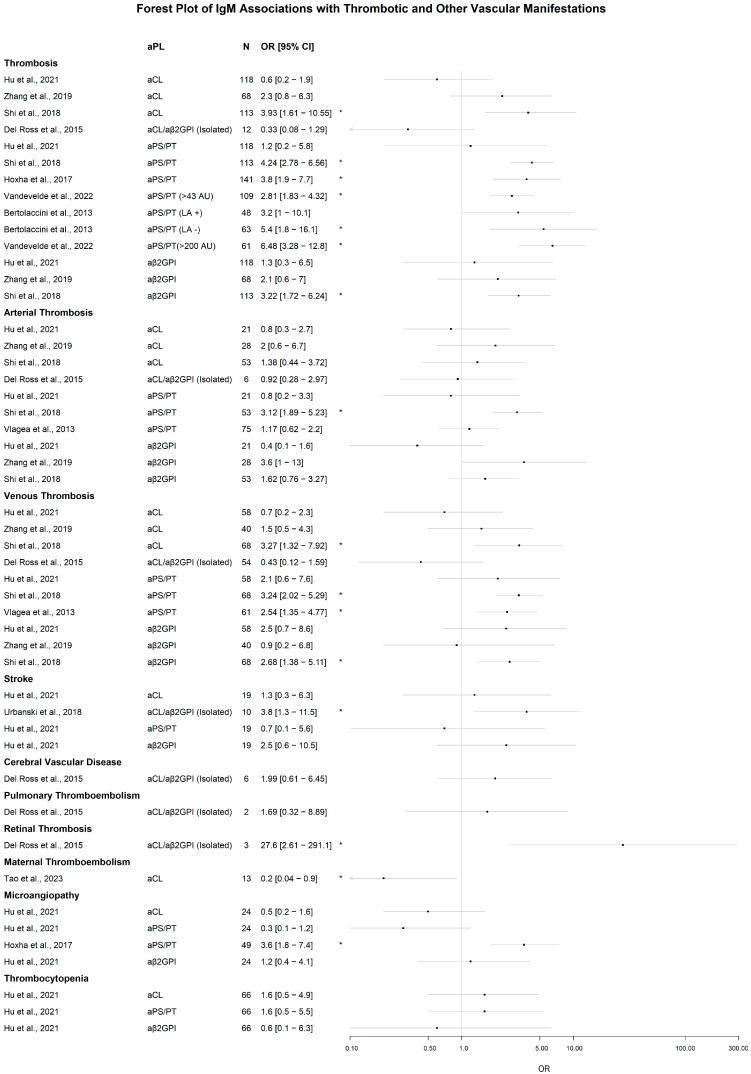
Forest plot of the odds ratios (ORs) of IgM aPL in different thrombotic and vascular clinical manifestations. Data are ordered by the aPL and represented as OR [95% confidence interval]. Significant results are marked with *. N represents the number of patients with the particular manifestation. In the case of isolated IgM, the N is manifestation and isotype-specific. The scale is logarithmic, aβ_2_GPI: anti-β_2_-glycoprotein I; aCL: anti-cardiolipin; aPS/PT: anti-phosphatidylserine/prothrombin; AU: arbitrary units; LA: lupus anticoagulant; +: positive; −: negative. Cited works: Hu et al. [61], Zhang et al. [66], Shi et al. [42], Hoxha et al. [43], Vandevelde et al. [50], Del Ross et al. [51], Bertolaccini et al. [65], Vlagea et al. [45], Urbanski et al. [59], Tao et al. [63].

## Data Availability

All data generated or analysed during this study are included in the respective cited articles.

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
