# Peer review of "IgM Antiphospholipid Antibodies in Antiphospholipid Syndrome: Prevalence, Clinical Associations, and Diagnostic Implications—A Scoping Review"

_jcm, 2025, doi:10.3390/jcm14207164_

Round 1

Reviewer 1 Report

Comments and Suggestions for Authors

The manuscript entitled "IgM Antiphospholipid Antibodies in Antiphospholipid Syndrome: Prevalence, Clinical Associations, and Diagnostic Implications— A Scoping Review" in which the authors synthesise evidence from recent literature on the prevalence, associations, and clinical value of IgM isotypes of aCL, aβ₂GPI, and aPS/PT antibodies across various APS phenotypes, including obstetric and thrombotic subsets, paediatric cases, and non-criteria manifestations. They found that the role of aPL-IgM remains uncertain.

The work is understandable. However, this paper suffers from some shortcomings that need modification.

Shortcomings:

  • In Introduction section,
  • Please add the risk factors and the complications of Antiphospholipid syndrome.
  • Please add the criteria of diagnosis of Antiphospholipid syndrome.
  • In Discussion section,
  • Please discuss in detail, the role of IgM antiphospholipid antibodies in patients with Antiphospholipid syndrome including the beneficial and detrimental effects in those patients.
  • Please discuss in brief, the therapeutic agents that could target IgM antiphospholipid antibodies in patients with Antiphospholipid syndrome.

4- Please add your future perspectives.

Author Response

Comment 1: In the Introduction, please add the risk factors and the complications of Antiphospholipid syndrome. 
Response 1: Thank you for this suggestion. We have added a brief overview of established cardiovascular and thrombotic risk factors to Introduction as follows:  

“Risk factors in APS are the presence of LA, double/triple positivity in high titres, concomitant SLE, classical cardiovascular risk factors such as hypertension, dyslipidaemia, obesity, smoking, and diabetes [10,11].” 

Further, we summarised the major complications associated with APS, with supporting citations (paragraph 1). We also summarised the major APS complications with citations. To avoid redundancy, we refer readers to Figure 1, where these are implicit in the classification framework. Specifically: 

“These include venous thromboembolism, arterial thrombosis, microvascular diseases, thrombocytopenia, cardiac valve affections, and recurrent miscarriages, foetal death and placental vasculopathies [4,5].” 

Comment 2: In the Introduction, please add the criteria of diagnosis of Antiphospholipid syndrome. 
Response 2: We appreciate this comment and agree that including classification criteria enhances the comprehensibility of the work. We have now added a concise summary of the current APS classification criteria (2023 ACR/EULAR) in Figure 1, and the legend has been expanded accordingly. We also added more information on the weighted scoring system and its interpretation to the Figure 1 legend and additional text to paragraph 3. As true diagnostic criteria do not yet exist, we clarified this distinction and referenced ongoing efforts, including the work by Vandevelde et al. (2024) and Arachchillage et al. (2024) on harmonised interpretation of the criteria for diagnostic application. 

Comment 3: In the Discussion, please discuss in detail the role of IgM antiphospholipid antibodies in patients with Antiphospholipid syndrome, including beneficial and detrimental effects. 
Response 3: Thank you for this valuable comment. We have substantially revised the Discussion to place greater emphasis on the potential role of IgM aPL, highlighting both detrimental associations (e.g., risk of thrombosis or adverse pregnancy outcomes in isolated IgM phenotypes) and possible modulatory or protective effects observed in mechanistic studies. We also discussed the limitations in current evidence—particularly the small number of isolated IgM cases and lack of consistent per-patient stratification—that complicate interpretation. The Results and Discussion sections have undergone major changes as requested by the reviewers. To address your point specifically, we changed the section explaining significant positive associations with OR >1 and inverse associations with OR<1. Specifically: 

“In our review, positive associations appeared across several obstetric endpoints, with aCL-IgM showing the most frequent positive signals across pregnancy loss and placenta-mediated complications. Some complications appeared as significantly not-associated with aβ₂GPI-IgM (OR < 1). We interpret these inverse findings as cohort-specific artefacts, since in one key study, only a small fraction of women with obstetric complications were aPL-positive, reducing the power to detect true effects [35].” 

Comment 4: Please discuss in brief the therapeutic agents that could target IgM antiphospholipid antibodies in patients with Antiphospholipid syndrome. 
Response 4: We appreciate this point. While no therapy specifically targets IgM aPL, we have added a new paragraph in the Discussion outlining agents that indirectly influence aPL biology or downstream pathways. These include anticoagulation, low-dose aspirin, heparin (with some evidence of effects on aβ₂GPI-IgM), and other immunomodulatory agents, though none are IgM-selective. Although therapy was not the primary focus of this review, the additions clarify that no therapeutic agents specifically target IgM aPL. Specifically: 

“Current APS management is targeting symptomatology and is not directed against aPL isotypes. Long-term anticoagulation for TAPS and low-dose aspirin with or without heparin for OAPS remain the standards of care, with none of them specifically targeting IgM aPL [11]. Several agents may modulate aPL biology or downstream pathways, but evidence is indirect and not IgM-selective. Heparin interferes with aPL-mediated trophoblast and coagulation-surface interactions in experimental systems and improves pregnancy outcomes clinically, yet this is a mechanistic rather than isotype-specific effect [82,83]. In a recent study, heparin could lower aβ₂GPI-IgM levels while keeping aCL-IgM levels unaffected, but the molecular mechanism remains unknown [84]." 

Comment 5: Please add your future perspectives. 
Response 5: We thank the reviewer for this suggestion. A dedicated “Future Perspectives” section has been added after the Conclusion, emphasising the need for prospective, standardised studies with harmonised assays, patient-level IgM data, and mechanistic endpoints to clarify whether IgM aPL act as bystanders, phenotype-specific modifiers, or true pathogenic drivers. Specifically: 

“Future Perspectives  

To accurately determine the actual effect of IgM antibodies, prospective, standardised studies are essential. Ideally, incorporating uniform cut-offs, harmonised aPL data presentation stratified by isotype, including specific individual patient descriptions, and mechanistic endpoints such as NETosis, complement activation and placental histology. Future studies should prospectively track IgM-only cohorts to determine the clinical background and outcomes of this rare population. Only with this level of methodological rigour can we determine whether IgM antibodies are innocuous bystanders, phenotype-specific modifiers, or proper pathogenic drivers.” 

Reviewer 2 Report

Comments and Suggestions for Authors

The manuscript focuses on antiphospholipid antibodies of the IgM class and their role in thromboembolic episodes as well as other clinical manifestations.

The introduction provides a thorough description of antiphospholipid antibodies and includes the updated APS classification criteria. The methodology is excellent and clearly presented, supported by a well-designed flow chart. The results are structured according to multiple aspects, extending beyond thromboembolic events alone. The tables comprehensively present the distribution of APLA in patients. Figures 2 and 3 compare APLA across different studies, including odds ratios, which is very informative.

A particularly interesting section is the discussion on NETosis. The discussion overall is well-balanced, with no comments needed for improvement. The references are also appropriate and up to date.

Congratulations to the authors on preparing a very strong and valuable review.

Author Response

Comment 1: The introduction provides a thorough description of antiphospholipid antibodies and includes the updated APS classification criteria. The methodology is excellent and clearly presented, supported by a well-designed flow chart. The results are structured according to multiple aspects, extending beyond thromboembolic events alone. The tables comprehensively present the distribution of APLA in patients. Figures 2 and 3 compare APLA across different studies, including odds ratios, which is very informative. 

A particularly interesting section is the discussion on NETosis. The discussion overall is well-balanced, with no comments needed for improvement. The references are also appropriate and up to date. 

Congratulations to the authors on preparing a very strong and valuable review. 

Response 1: We thank the reviewer for their generous and encouraging assessment. Nonetheless, we have refined the manuscript to further improve its content and readability. First, we added a succinct summary of the 2023 ACR/EULAR classification criteria to the Introduction to better situate the debate around IgM. In parallel, we streamlined the Discussion—removing repetition, strengthening topic sentences, and focusing on the main take-home messages—so that the argument now reads more clearly from prevalence to clinical interpretation. We also clarified the presentation of significant findings: study-level signals are now highlighted more explicitly in Figures 3 and 4 (and their legends), thereby improving readability at a glance and alignment with the Results narrative. Finally, we incorporated a brief paragraph on therapeutic approaches within the Discussion, outlining current management and the extent to which it modulates IgM aPL. We trust these changes enhance clarity while preserving the strengths noted by the reviewer. 

Reviewer 3 Report

Comments and Suggestions for Authors

This is an interesting review on the role of IgM Antiphospholipid Antibodies in APS - a matter of current debate. Although the topic is of great interest there are some major limitations to this review:

  1. First and foremost, as the authors stated aPL of the IgM isotype were de-emphasized in the 2023 ACR/EULAR criteria if exclusively present. Hence the review should focused primarily on data relevant to the role of isolated IgM in regards to APS manifestations and outcomes. Notably, some data are presented however only in the discussion section (see 4.4.1 and 4.4.2).
  2. The vast majority of the review deals with the role of IgM in various studies in which all aPL were assessed. Although deciphering the data on IgM-aPL in general is of interest is has been described previously and should be concise and presented in a more "reader friendly manner" as currently it is difficult to follow. For concomitant IgM significance in purely OAPS, combined OAPT/TAPS and purely TAPS (3 sections will do)
  3. The plot graphs are nice (number of samples is of interest)
  4. The discussion is written as a "summary" of the data and should be re-edited. No need to repeat the study design etc, bur rather discuss the major points. On the other hand new data should not be presented for the first time in the discussion  

Author Response

Comment 1: 1. First and foremost, as the authors stated aPL of the IgM isotype were de-emphasized in the 2023 ACR/EULAR criteria if exclusively present. Hence the review should focused primarily on data relevant to the role of isolated IgM in regards to APS manifestations and outcomes. Notably, some data are presented however only in the discussion section (see 4.4.1 and 4.4.2).

Response 1: We thank the reviewer and strongly agree that the most insightful data would be from only isolated IgM cohorts. The limited number of cohorts reporting truly isolated IgM serology constrained the available evidence. Isolated IgM phenotype was previously untested and therefore unknown. Often, IgM/IgG aPL isotypes were reported together as under the previous classification criteria, they had the same weight. Only very recently, there has been a push to report and study isolated IgM, so only a few studies exist – earlier criteria did not incentivise separate reporting of isolated IgM, so only a few such studies exist. 

To address your comment, we highlighted the isolated IgM results in the Discussion and made sure no new data is presented in the Discussion. Here is a snippet of the part discussing isolated IgM data: 

“In thrombotic and vascular manifestations, the most informative data come from 3 studies of isolated IgM serology. aPS/PT-IgM showed a positive association with thrombosis [65], while isolated aCL/aβ₂GPI-IgM was linked to retinal occlusion [51] and in a separate cohort, to stroke [59], although sample sizes were small.” 

Comment 2: 2. The vast majority of the review deals with the role of IgM in various studies in which all aPL were assessed. Although deciphering the data on IgM-aPL in general is of interest is has been described previously and should be concise and presented in a more "reader friendly manner" as currently it is difficult to follow. For concomitant IgM significance in purely OAPS, combined OAPT/TAPS and purely TAPS (3 sections will do) 

Response 2: We appreciate the reviewer’s suggestion to structure the results into three sections focused on the significance of IgM in purely OAPS, combined OAPS/TAPS, and purely TAPS. We agree that this approach provides clarity and reflects the traditional clinical phenotypes that have been considered so far. The new ACR/EULAR classification criteria go beyond the purely obstetric or thrombotic phenotypes and consider a wider spectrum of clinical manifestations. For this reason, in addition to the proposed three-section structure, we have also emphasized the broader significance of IgM positivity in other clinical aspects, which we consider of particular relevance for a comprehensive understanding of APS. 

To implement your comment, the Results section has been widely revised and rewritten to improve clarity and comprehensibility of the text. 

Comment 3: 3. The plot graphs are nice (number of samples is of interest) 

Response 3: We appreciate the reviewer’s comment, and we agree that the size of a cohort is important in interpreting odds ratio values. We initially wanted to include the N, but were confronted with the fact that specific data on aPL positivity per manifestation is rarely available. Providing the total cohort N could be misleading, as aPL positivity often varies by manifestation. We have therefore added a column indicating the number of patients per manifestation where available. To address your comment, we added another column to the Figures 3 and 4, indicating the number of patients with a particular manifestation. And describe it in the Results section 3.2 like so: 

“The N value associated with each manifestation represents the number of patients with the given clinical manifestation. Only in the cases of isolated IgM aPL results, the N is manifestation- and aPL-specific.”

Comment 4: 4. The discussion is written as a "summary" of the data and should be re-edited. No need to repeat the study design etc, bur rather discuss the major points. On the other hand new data should not be presented for the first time in the discussion   

Response 4: We appreciate your feedback. To address your comment, we excluded repetitions of numerical values of Results in the Discussion. Further, we eliminated all unrelated IgM data and made the section more concise. Overall, the Discussion section underwent a considerate editing, thus must be viewed as a whole.

Reviewer 4 Report

Comments and Suggestions for Authors

This manuscript presents a thorough and comprehensive review of the prevalence and clinical significance of IgM antiphospholipid antibodies in antiphospholipid syndrome (APS). The key strength is in comparison of the role of IgM in thrombotic and obstetric manifestations of APS. The review offers valuable insights into the diagnostic and prognostic relevance of IgM aPLA  in clinical practice. There are only minor points to address:

  1. Explain in more detail the reasons for the full text records exclusion
  2. In the first paragraph of 3.1.5.Cerebrovascular manifestation, there is a typo in sentence “aPS/PT-IgM was present in 25.0% of LA-negative In contrast, in LA-negative patients, the aPS/PT-IgM prevalence surged to 85%. I believe, ne of them should be LA- positive. Please correct.
  3. It would be useful to mark significant associations in Figure 2 and
  4. The phrase “a series of studies did not report a statistically significant associations” doesn’t seem to be correct from the context. It implies that the significance was not reported. Clearer would be phrase “A series of studies found that the association was not statistically significant”.
Comments on the Quality of English Language

The English is generally fine, requires only minor improvement to convey the right message. 

Author Response

This manuscript presents a thorough and comprehensive review of the prevalence and clinical significance of IgM antiphospholipid antibodies in antiphospholipid syndrome (APS). The key strength is in comparison of the role of IgM in thrombotic and obstetric manifestations of APS. The review offers valuable insights into the diagnostic and prognostic relevance of IgM aPLA  in clinical practice. There are only minor points to address: 

Comment 1: 1. Explain in more detail the reasons for the full text records exclusion 

Response 1: We appreciate the reviewer’s comment and recognise that a more complete explanation on records exclusion will be helpful for the reader, therefore we included a more detailed explanation in the methods section like so: 

“Full text of 297 articles was assessed; 257 failed the eligibility criteria shown in Figure 1. For example, excluded studies were those that (i) included aPL data, but the cohort was made of patients with other diseases, such as SLE, (ii) had no IgM data or no IgM data separate from IgG, (iii) focused on comparing different detection strategies rather than APS diagnosis or pathology, (iv) included other aPL than those studied or (v) other reasons. The “Other” category pooled studies with <15 participants, in vitro studies, or guidelines.” 

Comment 2: 2. In the first paragraph of 3.1.5.Cerebrovascular manifestation, there is a typo in sentence “aPS/PT-IgM was present in 25.0% of LA-negative In contrast, in LA-negative patients, the aPS/PT-IgM prevalence surged to 85%. I believe, ne of them should be LA- positive. Please correct. 

Response 2: We thank the reviewer for spotting this error. This typo has been fixed, and the sentence is now clear:  

“ In an isolated aPS/PT stroke cohort, aPS/PT-IgM was detected in 85% of LA-negative versus in only 25% of LA-positive patients [58].” 

Comment 3: 3. It would be useful to mark significant associations in Figure 2 and 

Response 3: We thank the reviewer for this comment and agree that the results are more easily interpreted when the significant results are highlighted. Therefore, we added a significance column to Figures 3 and 4 (formely Figures 2 and 3) and all significant results are marked with an asterisk. 

Comment 4: 4. The phrase “a series of studies did not report a statistically significant associations” doesn’t seem to be correct from the context. It implies that the significance was not reported. Clearer would be phrase “A series of studies found that the association was not statistically significant”. 

Response 4: We appreciate the reviewer’s comment and agree that the wording we chose indicated an unintended meaning. To address this, we rephrased the sentence to convey the message that we intended, like so: 

“A series of studies reported insignificant associations of at least one IgM aCL, aβ₂GPI or aPS/PT positivity with pregnancy loss ([35,45,61]), embryonic loss ([62]), ≥ three consecutive miscarriages at < 10 WG ([35,64]), foetal death at > 10 WG ([35,64]), premature birth < 34 WG ([35,64]), early-onset PE ([30,63,64]), PE ([30,62]), foetal growth restriction ([62,64]), and SGA neonate ([30] for aβ₂GPI).” 

Round 2

Reviewer 3 Report

Comments and Suggestions for Authors

no further comments